# Safeguarding adolescent mental health in India (SAMA): study protocol for codesign and feasibility study of a school systems intervention targeting adolescent anxiety and depression in India

Siobhan Hugh-Jones [1] N Janardhana,[2] Hareth Al-Janabi,[3] Poornima Bhola,[2] Paul Cooke,[4] Mina Fazel,[5] Kristian Hudson,[6] Prachi Khandeparkar,[7] Tolib Mirzoev [8] Surendran Venkataraman,[9] Robert M West [10] Pavan Mallikarjun[11]

For numbered affiliations see end of article.

**Correspondence to**
Dr Siobhan Hugh-Jones;
s.hugh-jones@leeds.ac.uk

## ABSTRACT

**Introduction** Symptoms of anxiety and depression in Indian adolescents are common. Schools can be opportune sites for delivery of mental health interventions. India, however, is without a evidence-based and integrated whole-school mental health approach. This article describes the study design for the safeguarding adolescent mental health in India (SAMA) project. The aim of SAMA is to codesign and feasibility test a suite of multicomponent interventions for mental health across the intersecting systems of adolescents, schools, families and their local communities in India.

**Methods and analysis** Our project will codesign and feasibility test four interventions to run in parallel in eight schools (three assigned to waitlist) in Bengaluru and Kolar in Karnataka, India. The primary aim is to reduce the prevalence of adolescent anxiety and depression. Codesign of interventions will build on existing evidence and resources. Interventions for adolescents at school will be universal, incorporating curriculum and social components. Interventions for parents and teachers will target mental health literacy, and also for teachers, training in positive behaviour practices. Intervention in the school community will target school climate to improve student mental health literacy and care. Intervention for the wider community will be via adolescent-led films and social media. We will generate intervention cost estimates, test outcome measures and identify pathways to increase policy action on the evidence.

**Ethics and dissemination** Ethical approval has been granted by the National Institute of Mental Health Neurosciences Research Ethics Committee (NIMHANS/26th IEC (Behv Sc Div/2020/2021)) and the University of Leeds School of Psychology Research Ethics Committee (PSYC-221). Certain data will be available on a data sharing site. Findings will be disseminated via peer-reviewed journals and conferences.

## Strengths and limitations of this study

► A key strength of this study is its focus on depression and anxiety prevention, which are common mental health conditions among Indian adolescents.
► The study uses a public mental health systems approach, which adopts a 'person-in-context' perspective and recognises multiple determinants of mental health.
► Extensive codesign of interventions will promote cultural relevance and acceptability by key stakeholders, including adolescents, teachers, parents and school communities.
► The COVID-19 pandemic makes this research more pertinent but may affect recruitment of the schools/adolescents or may lead to participation bias if the study processes are conducted online by excluding those with limited access or who lack technical skills.
► The opt-in approach to consent may lead to participation bias.

## INTRODUCTION

Mental health conditions including depression and anxiety disorders are among the top 10 causes of illness and disability in adolescents.[1] Half of all lifelong mental health conditions have their onset in adolescence.[2] Poor mental health in adolescence is associated with poor physical health and lifelong disadvantage, especially if education is disrupted, impacting occupational and social trajectories.[3]

India has the largest adolescent population in the world (253 million).[4] An estimated 9.8 million Indian aged 13–17 year

olds (pooled prevalence of 7.3%) have a clinical mental health condition[5] but levels seem considerably higher in urban areas and among school-going adolescents.[6] Although there are limited data on the treatment gap in adolescents, the overall treatment gap for mental disorders in India is around 90%.[5] The treatment gap and the effectiveness and cost-effectiveness of early intervention are driving investment into public mental health in efforts to develop scalable programmes that can reach adolescents before their symptoms progress to levels of a clinical disorder.[5] Tackling preventable and treatable health conditions, and improving the quality of education for adolescents, present the single best investments for health and well-being a low-income and middle-income country (LMIC) can make.[7]

Schools have tremendous potential as platforms for public mental health interventions, which can secure positive effects on mental health, including a reduction in anxiety and depression, and which can be delivered, sustained and scaled in LMICs.[7] School provision can increase access to interventions over and above existing mental healthcare systems, which in LMICs, are typically fragmented, under-resourced and not tailored to adolescents.[8–10] Improving school experience may also promote longer participation in education, which reduces a number of risks for girls in particular.[8] The Lancet commission on adolescent well-being calls for investment in school mental health, bolstered by the rise in school attendance[8] including in India (97% enrolment).[11] Although some adolescent mental health programmes exist in India, there is as yet no integrated, evidence-based, whole-school mental health approach for adolescents.

Our project responds to this need by identifying existing school interventions, which can be integrated, tailored and culturally adapted into a systems approach[12] in India through a process of coproduction with stakeholders. Given the multiple determinants of adolescent mental health in India,[5] interventions will need to target individual and contextual factors. In India, a key contextual determinant of adolescent mental health is the nature of schooling, including the use of corporal punishment,[13–15] extreme academic pressure,[16 17] a lack of mental health literacy among adolescents, teachers and parents,[18–20] prevailing mental health stigma and bullying,[21] poorly supported staff, often with large class sizes and a lack of mental health support within schools.[22] Although the Government of India endorsed the WHO Health Promoting School model (2007),[23] progress on school mental health has been slow. To date, school initiatives in India have mostly targeted physical health and life skills.[24–27] There is an urgent need to accelerate improvements in mental health in Indian schools, from a safety and rights perspective,[28] to reduce school risks to adolescent mental health and to improve population health. India has limited evidence-based, scalable school programmes, endorsed by policy, to support adolescent mental health. This is a critical care gap.[5 29] Our study

aims to contribute to the evidence base about what works in Indian schools to support adolescent mental health.

There is some evidence for school-based universal mental interventions for depression and anxiety prevention, and mental health promotion, in high-income countries.[30 31] The evidence in LMICs is promising though limited.[8–10] Three whole-school health and life-skills programmes with small mental health components have been tested in India.[24 26 32 33] One did not report any beneficial clinical outcomes[24] and one has yet to establish effectiveness,[32] indicating that life-skills interventions may be insufficient to impact adolescent anxiety and depression. The third is a health promotion programme (multicomponent school health promotion programme (SEHER)) focusing on school climate, and targeting physical and sexual health, bullying, gender equality and depressive symptoms. It focused on promoting adolescents' social and problem-solving skills, engaging adolescents, teachers, and parents in school-level decision-making and delivering factual knowledge about health and risk behaviours to the school community.[26] Outcomes were positive, including for depressive symptoms, with benefits extended up to two years.[33] It did not, however, target anxiety nor include broad mental health promotion. This is a critical gap given the prevalence of anxiety disorders in adolescence and its comorbidity with depression.

Our study, safeguarding adolescent mental health in India (SAMA), builds on learning from SEHER in terms of the value and acceptability of whole-school, multicomponent interventions in Indian schools and the contribution of school climate to adolescent well-being. Project SAMA extends SEHER in several ways. Our health objective is to reduce symptoms of both anxiety and depression in symptomatic adolescents and to promote the well-being all school-going adolescents. We will codesign, and feasibility test, a suite of school interventions in India targeting multiple risks and protective factors across the school system. We extend SEHER by including codesigned interventions targeting teachers and parents, as well as an adolescent mental health psychoeducation intervention. We also augment the school climate intervention of SEHER via coproduction. The SAMA study will take place between January 2021 and December 2023.

## METHOD AND ANALYSIS
### Study design
SAMA will run over 36 months and will be delivered via eight work packages (WP) (see figure 1).

WP1–4 are intervention codesign/coadaptation and feasibility studies following Medical Research Council guidance[34] and the Six Steps in Quality Intervention Development,[35] targeting adolescents (14–15 years), teachers, the school community and parents (table 1). WP5 is implementation research on these interventions and WP6 is an exploratory cost-effectiveness study of the interventions. WP7 and 8 target policy and local community systems, respectively. Year 1 will focus on codesign

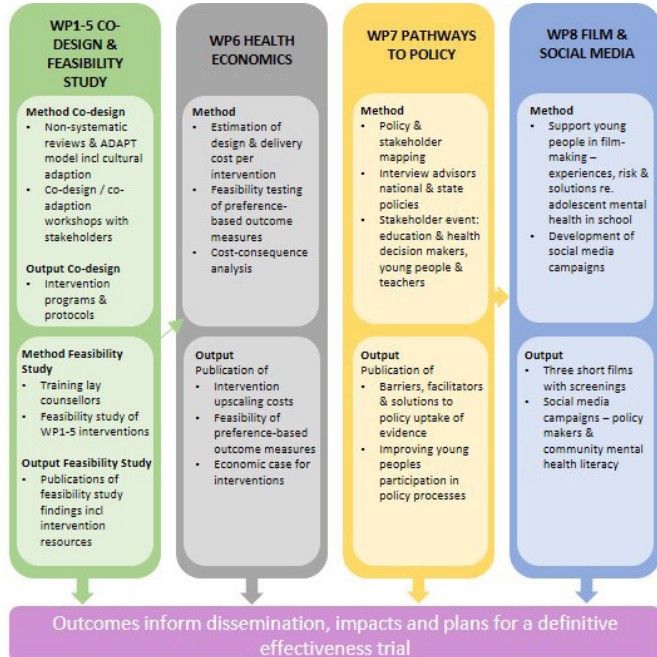

**Figure 1** Study work packages (WP).

and coadaptation of interventions via WP1–5. Years 2 and 3 will include a feasibility study of the adapted interventions alongside WP6 and 7. WP8 will run across all 3 years.

SAMA will work with adolescents as partners in all stages of the project. In addition to the adolescents involved as participants in codesign, we will have a Youth Advisory Board who will be consulted on all project components, a subset of whom will form adolescent film crews and drive the social media campaigning (both WP8). We will have three other advisory boards (Steering Group, Scientific Advisory Board, Ethics Oversight Committee).

For WP1-4, some broad interventions requirements have been defined in advance of codesign, including the aims and primary and secondary outcomes. These are shown in table 1. It is possible that codesign discussions may lead to the inclusion of additional outcome measures. All interventions will be delivered by external lay counsellors. Use of lay counsellors can represent effective task-shifting task, an approach endorsed by the WHO in low-resource settings.[36] Task shifting involves assigning

tasks and responsibilities from more specialised to less specialised groups and requires clearly defined roles with effective training and supervision. There is some evidence that this delivery approach is superior and preferred to teacher-delivered interventions, and peer educator programmes, at least for adolescent mental health programmes.[22 26 32] Notably, the cluster randomised trial of SEHER conducted across 75 schools in Goa was found to be effective when delivered by lay counsellors but not teachers.[26] There are additional benefits of having a small team of lay counsellors working closely in partnership with schools. These include each lay counsellors having a co-counsellor to work alongside to deliver SAMA, counsellors being able to get to know particular schools well (ie, their systems, culture), having constancy in the lay counsellors so school staff and students can build familiarity and trust, and having a defined conduit for communication about SAMA within schools and between school and the study team.

### School recruitment
We will recruit eight schools in Kolar and Bangalore to participate in both the codesign and feasibility stages of SAMA (WP1–6). School recruitment will be facilitated by the Karnataka State Primary and Secondary Education Ministry and the Karnataka State Secondary School Headmaster's Association. We will recruit two school types (government and low-cost private) to identify school factors that may affect feasibility. Schools will be eligible to take part in SAMA if they endorse the delivery of WP1–6 in their schools, including participation in codesign. WP1 intervention feasibility testing will be with a new grade 9 cohort not involved in codesign.

### Year 1: codesign/coadaptation of interventions
Codesign will proceed in stages (see figure 2) and will include coadaptation of existing interventions as well as design of new components where needed (eg, content, implementation, safeguarding). We begin with coadapting interventions before identifying the need for new codesigned elements. Our approach to coadaptation is informed by phases 1–3 of the ADAPT model for introducing complex public health interventions into

| Table 1 | Intervention aims and requirements defined in advance | | | |
|---|---|---|---|---|
| **WP** | **Intervention** | **Requirements defined prior to codesign** | **Primary outcome** | **Secondary outcomes (include but not restricted to)** |
| 1 | Whole-school curriculum adolescent mental health programme | 15–16 year olds (grade 9) as advised by schools 6–8-week duration | Symptoms of anxiety and depression | Mental health literacy, stress, self-esteem, resilience, quality of life, school attendance and performance |
| 2 | Teacher education programme | Low intensity Suitable for all levels of staff | Teacher mental health literacy | Teacher attitude towards and use of positive behaviour management practices |
| 3 | School climate intervention | Up to 6 months | School climate | Mental health literacy, perceived stigma, perceived support |
| 4 | Parent mental health literacy intervention | Low intensity | Parent mental health literacy | Requests for more mental health information/support/referrals |

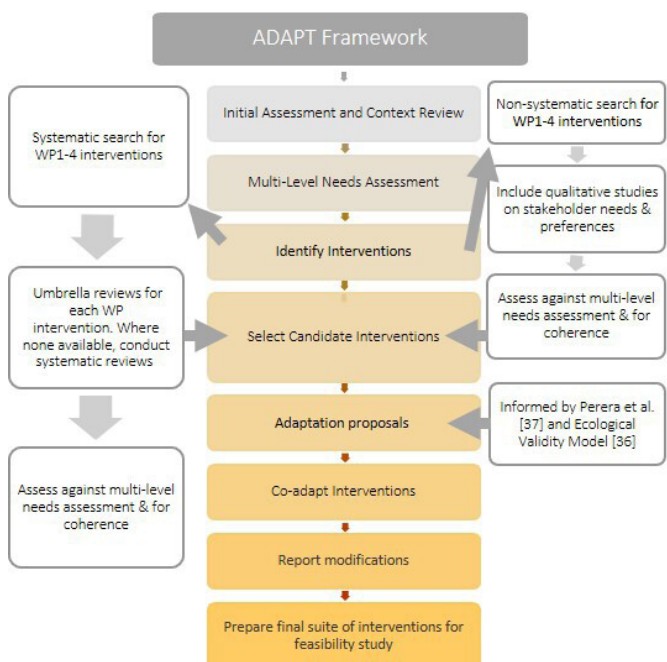

**Figure 2** Stages of coadaptation/codesign. WP, work packages.

new settings[37] and also by Perera *et al.*'s approach[38] to adapting low-intensity psychological interventions into new cultural settings. WP5 runs alongside WP1–4 and aims to identify implementation facilitators and barriers for all intervention components. As well as adapting interventions, we will coproduce with adolescent and stakeholders a whole-school safeguarding protocol tailored to the interventions.

### Phase 1: exploration of needs and existing interventions
#### Initial assessment
A desk review will be conducted to gather knowledge on the local context and research evidence relevant to our project aims, including identification of any existing best practice and policy requirements. This will feed into an evidence-based, multilevel needs assessment of endusers, families, schools and regional/state systems to sensitise our selection of interventions and implementation approach.

#### Intervention selection and exploration
We aim to identify interventions that are likely to be acceptable, feasible and efficacious in our target context to meet our project aims. We will search for interventions with the potential (after coadaptation) to impact at least the primary outcome for each WP (table 1). As per figure 2, a non-systematic search will be conducted first to collate or identify interventions which are known to the team in India, referenced in Indian education or mental health policies or referenced by Indian or global organisations (eg, Save the Children, WHO). This will be followed by a systematic search. Umbrella review protocols for these are registered on PROSPERO[39–42] and reviews are nearing completion. Considering search returns from both the

non-systematic and systematic searched, WP1–4 leads will propose candidate interventions to be taken to Phase 2. While established effectiveness of an intervention is important, we also need to consider the coherence, integration, implementation and scalability of interventions for the local context.[43] Proposals will be reviewed by the research team and project advisory boards, which include in-country adolescents, schools and mental health professionals as well as the team which led SEHER.[26] Amendments to proposals will be made where necessary.

This phase includes work on implementation protocols (WP5). We will build on a UK framework for a whole-school mental health intervention[44] based on the Consolidated Framework for Implementation Research (CFIR).[45] The framework will be iteratively developed as SAMA interventions are finalised, drawing on existing implementation evidence from India, including the SEHER study,[26] and through our coadaptation processes.

### Phase 2: preparation for codesign
#### Identification of mismatches and model development
Candidate interventions will be assessed against our multilevel needs assessment; those failing to meet significant need will be rejected. We will then develop a model of how the final interventions could work across systems before identifying the aspects (across surface and deep levels[46]), which are most critical to take forward to codesign/coadaptation. Decisions here will be informed by the approach of Perera *et al.*[38] and their use of the Ecological Validity Model.[47] This will lead to the generation of adaptation hypotheses, that is, our proposals for which intervention aspects will need to be culturally adapted. The Ecological Validity Model prompts direct consideration of the language, persons, metaphors, content, concepts, goals, methods and context (which includes implementation) of the intervention to be adapted. We will supplement these proposals with codesign needs, that is, new features which need to be created because of the systems approach or local school context. Coadaptation and codesign proposals will be finalised in consultation with our youth and advisory boards.

#### Coadaptation and codesign
Our aim in this stage is to optimise the acceptability, feasibility, efficacy and meaningful evaluation of the suite of interventions by involving endusers and stakeholders in intervention adaptation and its associated implementation approach. We will host codesign/coadaptation workshops for each WP intervention, each involving adolescents (14–16 year olds, single-gendered groups with more females), parents, schools (including teachers and institute heads) and key stakeholders in separate workshops. WP2 codesign will additionally involve national organisations (eg, State Education & Health Department, Teachers of India, ChildLine India, and Save the Children). WP4 codesign will begin with a stakeholder event to identify community-led solutions to reaching parents. Creative evaluation methods will be devised during

codesign to be sensitive to low literacy levels, low chance of measure completion and desirability bias. Implementation barriers, facilitators and developing implementation protocol will be a standing item for consultation in codesign workshops. Codesign/coadaptation workshops will include a range of age-appropriate and role-appropriate discussions and activities centering on the coadaptation proposals developed in the previous stage.

### Phase 3: undertaking modifications

We will modify the interventions based on coadaptation and codesign workshop outputs. We will use the expanded Framework for Reporting Adaptations and Modifications-Expanded (FRAME) approach to report modifications to interventions.[48] This approach documents when, why and how modifications were made, and differentiates cultural adaptation from adaptations made for other reasons. Proposed modifications will be reviewed by the full team to ensure modified interventions still meet the multilevel needs assessment, retain active ingredients and are coherent as a suite of interventions. Modifications and final prototype interventions will be reviewed by our youth and advisory boards as well as a board of in-country mental health professionals and education sector representatives to ensure they meet state and national requirements. Once finalised, we will produce for each WP intervention a logic model, programme manual, intervention resources, lay counsellor training and supervision protocol, implementation and evaluation protocols.

### Year 2: feasibility testing

Lay counsellors will be from the local areas, have at least a master's qualification in psychology or a related discipline and experience in working with young people and/or schools. Prior to the launch of the feasibility study, they will receive full training over 3 months by a clinical psychologist on the team, supported by another team member experienced in training lay counsellors. Trainees will deliver multiple mock sessions of intervention delivery to support readiness for delivery in schools and quality control. The feasibility study will run in eight schools, with three randomly assigned to waitlist. The four interventions will run in parallel in schools. Lay counsellors will complete weekly monitoring logs of intervention delivery in each school, including logging of modifications made (as per FRAME[48]) and will have weekly supervision by clinical team members. Fidelity monitoring will

be undertaken by a team of trained data collection assistants who have been heavily involved in the intervention development stage and will be highly familiar with the implementation protocol. Fidelity monitoring reports will be discussed frequently with the research manager to identify needs for lay counsellor action/support or intervention modification (see WP5).

We have established preliminary feasibility indicators for each intervention, which will be updated when interventions are finalised. Feasibility indicators which will be common across WP1–4 interventions are: meeting recruitment and retention rates, delivery as intended, measure completion and no significant adverse effects. We will adopt a 'traffic light' system for recruitment and retention as follows: red—if rates below 50% (say) then will not proceed; amber—if rates 50%–65% then proceed if there are plans to improve; green—above 65% proceed to definitive trial. Mixed-methods process and evaluation data will be secured for all interventions from adolescents, teachers, parents and lay counsellors to explore: perceived usefulness of the interventions; implementation challenges; unintended negative effects; safeguarding challenges; the process of change and intervention development needs. Evaluation outcomes will inform intervention improvements and parameters for a definitive trial.

### Sample size calculation

The planned sample for the feasibility study by intervention is detailed in table 2. The minimum recommended sample size for a feasibility study is 30.[49] The smallest group of participants are teachers (for the WP2 intervention). To secure a sample of 30, we will need to recruit a minimum of 8 schools, from which we will then be able to recruit approximately n=960 grade 9 adolescents (for WP1 intervention). Given this adolescent sample size, we have been able to calculate power in relation to WP1, where a key aim is to test two candidate primary outcomes measures—the Strengths and Difficulties Questionnaire (SDQ)[50] and the Revised Child Anxiety and Depression Scale (RCADS).[51] SDQ is an emotional and behavioural screening tool, culturally validated in India and used widely in adolescent mental health research. RCADS measures anxiety and depression and demonstrates excellent sensitivity to change. A translated RCADS has been used with Indian populations[52]

| WP | Intervention | N | Sample |
|---|---|---|---|
| **Table 2** | Planned sample size for each intervention in the feasibility study | | |
| 1 | Universal curriculum adolescent mental health programme | 960 | Grade 9 15–16 year olds from 8 schools (120 per school) |
| 2 | Teacher education programme | 32 | Grade 9 teachers (4 per school) |
| 3 | School climate intervention | 8 schools | To be defined following intervention selection |
| 4 | Parental mental literacy intervention | 960 sets of parents | Aim to reach all parents/carers of WP1 participants |

WP, work packages.

but full cultural validity needs to be established. As this will be an important tool to strengthen Indian mental health research, we will conduct a full validation study in parallel to the codesign stage. We have determined via a UK school feasibility trial[53] that the change in SDQ had a SD of 2.8 points. Assuming 80% recruitment, comparing the mean changes in SDQ from 480 participants from the intervention group with 288 from the waitlist group using a 1% significance level has more than 98% power to detect a difference of 1 point on the SDQ Scale. Around 960 adolescents will be approached for WP1. At 65% recruitment, the power remains over 95%. This calculation is based on an analysis of variance approach, that is, regressing the change in SDQ on the intervention status (intervention or waiting list). The power achieved from an analysis of covariance approach, which regresses the final SDQ Score on baseline and intervention status, should be similar. Should the mechanism of the WP1 intervention be similar across population subgroups, we can therefore show good power. There may, however, be differences in the operation of the intervention between subgroups, such as government versus private schools. In that case, assuming an even division of samples, the power becomes 85% with 80% recruitment and 78% with 65% recruitment. A significance level of 1% is used rather than 5% since the sensitivity of the outcome measures is sought to be compared—that is, the SDQ will be compared in sensitivity in this setting with the RCADS.

## WP5: implementation research

We will test the feasibility of our codesigned implementation approach, which will draw on improvement science. Lay counsellors will be trained and supported to optimise the intervention in situ by logging and overcoming implementation challenges as they arise, while keeping fidelity to intervention function. Process evaluations based on interviews/focus groups and implementation assessment tools will be conducted post intervention with key school staff and other stakeholder as needed depending on what challenges arise. Solutions to implementation challenges will be sought.

## WP6: health economic evaluation

Using a micro (bottom-up) costing strategy,[54] separate design and delivery costs per intervention component will be estimated using available data sources (eg, government ministry data, national pay scales, national earnings surveys) to determine unit costs for personnel time. Any additional primary data needs will be identified. Costs will be presented at the class and school level to promote generalisability. Preference-based outcome health and well-being measures are needed for economic evaluation[55] but there is limited evidence on their use with adolescents and no evidence of their use feasibility in this context (with adolescents in school-based settings and in India). We will feasibility test candidate measures including Child Health Utility (CHU)-9D[56] and EQ-5D-Y[57] via pre–post administration to 50 intervention adolescents and via 10

think-aloud interviews to identify completion challenges (a satisfactory sample size given the richness of interview data). We will also administer the EQ-5D-5L[58] and the ICEpop CAPability measure for Adults (ICECAP-A)[59] pre and post intervention to 50 parents to determine measure completion and any need for instrument adaptation. These sample sizes are not selected to estimate the significance of a specific outcome, but to balance the desire to not overburden individuals while still informing completion rates of instruments and items. We will use the cost and outcome data to generate exploratory cost–consequence analyses of the intervention strategies.

## WP7: understanding barriers and opportunities for research to policy uptake

We aim to (1) understand facilitators, barriers and solutions to policy uptake of the evidence on school mental health (in Karnataka); (2) identify ways of increasing adolescents' participation in policy processes and (3) identify the potential for application of solutions to other states. A policy and stakeholder mapping exercise will be undertaken, and then key advisors (n~10) for each of the three policies (two national and one state)[29] will be purposefully identified and interviewed. Insights will be shared at a stakeholder event comprising about 20 education and health decision-makers, adolescents and teachers to generate solutions for policy improvement. Insights will direct a social media campaign for policy action. Evaluation of effectiveness will be based on the number and nature of stakeholders reached, new knowledge on evidence-to-policy barriers and solutions and adolescents' participation.

## WP8: community film making and social media campaign

Around 15 adolescents (single-gendered groups and more females) will be supported to create three short films on experiences and risks around adolescent mental health, the potential for the school to affect mental health and solutions to safeguard their well-being. Films will be used in WP1–4 for engagement, education, advocacy and impact and to inform (1) a community and stakeholder's event. Films will be used in WP1–4 for engagement, education, advocacy and impact, via (1) a community and stakeholder's event to screen the films to promote shared understanding and solutions; (2) a social media education campaign targeting adolescents' mental health literacy; and (3) an advocacy campaign targeting national-level policy-makers. Outcomes will include open access films and new knowledge on the feasibility and reach of community filmmaking for health education and advocacy campaigning for school mental health in India.

## Data analysis plan
### WP1–5: Codesign/coadaptation

Workshop data will include audio/video recordings, facilitator notes and design materials (eg, maps, user journeys, reference ranking, annotated implementation and safeguarding protocols). Key workshop moments which

informed an adaptation or new design component will be transcribed, supplemented with facilitator notes and accuracy checked by two facilitators. Workshop data from different stakeholder groups per focus (eg, intervention, implementation, safeguarding) will be integrated by WP leads. They will synthesise proposed modifications and report them as per FRAME[48] to take forward to final approval stages in preparation for the feasibility study (described above).

### WP1–5: feasibility study

School-level effects will be handled as fixed rather than random effects. Primary analysis will be analysis of covariance where outcomes at 3 months are regressed on the baseline value and any relevant covariates/factors as age, sex, school, participant characteristics. The analysis will be supported by a two-level mixed-effects model where measures at two timepoints are nested within participants. If further to this analysis, a model is required to examine the relationship between the change in measure over time and the initial measure level, then Bloomqvist's method will be employed. Process evaluations will be analysed using framework analysis[60] or thematic analysis.[61] The implementation achievements in each school will be quantified per intervention and as a systems approach, and actions associated with success will be identified, drawing on the CFIR[45] and as per Hudson *et al.*[44]

### WP6: health economic evaluation

For the quantitative validity testing of the outcomes measures, we will calculate completion rates of the measures and items. For the qualitative validity testing, we will code the think-aloud interview transcripts to identify errors or difficulties in responding to items. Mean pre and post quality of life scores will be used to estimate quality of life benefits from the interventions. These data will be combined with estimates of the intervention costs in a simple cost–consequence analysis.[62]

### WP7: understanding barriers and opportunities for research to policy uptake

Data from the stakeholder analysis and document reviews will be continuously triangulated with analysis of transcripts from the in-depth interviews. All qualitative data will be analysed using framework approach[58] and guided by the three questions for WP7 and relevant theoretical frameworks for evidence-informed policy-making.[63–66]

### Patient and public involvement

Throughout SAMA, listening to adolescents will be central and incorporated in each WP. There are many barriers to adolescent involvement in low resource settings.[31] These can include high levels of mental health stigma as well as cultural differences where adolescents' opinions might be less commonly sought and heard. Our approach to adolescent participation will be informed by the WHO Global Consensus Statement on Meaningful Adolescent & Youth Engagement.[67] Adolescents will be trained and supported to contribute to the project. As well as being involved as participants in the design and evaluation of interventions in WP1–5, other adolescents will sit our Youth Advisory Board to act as 'critical friends' of the project. Adolescent contribution will also be via participatory filmmaking. This is a key platform within and beyond our project for youth voice in India on adolescent mental health and the role of schools in well-being. Other stakeholders including parents, teachers, the school community and policy-makers will be involved in the codesign and evaluation stages of the study.

### Ethics and dissemination

This research is being conducted in community settings and has received ethical approval from the National Institute of Mental Health Neurosciences Research Ethics Committee (NIMHANS/26th IEC (Behv Sc Div/2020/2021)), and the University of Leeds School of Psychology Research Ethics Committee (PSYC-221). Additional approvals have been granted by the Karnataka state primary and secondary education ministry and the Karnataka Department of State Education Research and Training.

We will seek consent from parents, teachers, school staff and other stakeholders and assent from adolescents. Grade 9 pupils will receive WP1 and 3 but can opt out of data collection. This is to ensure that participants' rights are not undermined by universal delivery. Teachers can access WP2 training without consenting to data collection. This is in recognition of the sensitivity of teachers reporting their mental health knowledge or use of harsh discipline practices, and responses will inform our understanding of the acceptability of our data collection measures. Audio/video recording in workshops and for WP8 will require explicit consent. Research data will be kept secure and confidential, with agreed UK–India data management plans and data sharing agreements.

Alongside WP1–5, we will be drafting, and refining through codesign, a stepped response and referral protocol to be assessed as part of the feasibility study. This protocol will draw on national and international guidance, as well as established practice in NIMHANS. Our priorities are response protocols for identification of (1) moderate-to-serious mental health need or risk (especially suicide risk) and (2) abuse (especially sexual abuse). The iterative production and field testing of this school mental health safeguarding policy will be an important contribution to the research field as well as to school systems.

Our plans for dissemination include: (1) events—launch, film screening, impact and capacity building events; (2) website—project website with multiple functionalities to engage, share resources, disseminate findings and in ways that can be sustained beyond the project; (3) social media (Twitter and Instagram) to reach multiple sectors and adolescents; (4) project briefs/updates/newsletters (flyers), especially for schools and community organisations; (5) executive reports to key stakeholders (eg, education and health departments, teacher training

colleges, mental health service providers); (6) sharing the project on global repositories, such as the Mental Health Innovation Network; (7) using existing communication platforms from University of Leeds Global Challenges projects relevant to adolescent well-being in LMICs; and (8) conferences and peer-reviewed publications.

## Strengths and limitations

A key strength of the study is our systems approach.[12] We target multiple risk and protective factors in the school setting as mental health is determined by both individual factors and the circumstances in which adolescents are living their day-to-day lives. A further positive about the study is its focus on anxiety and depression, which are the most common adolescent mental health conditions in India.[5 6] The likely feasibility and efficacy of the SAMA interventions is strengthened by several aspects of study design. Youth voice is central with the use of Youth Advisory Board and filmmaking. We combine learning from SEHER[26 33] with umbrella and systematic reviews of existing school interventions. Extensive investment is planned in intervention coadaptation/codesign with adolescents, schools, parents and key stakeholders, guided by established frameworks, to promote cultural relevance, ownership and acceptability.[37 38] Our whole-school approach fosters school ownership, is inclusive, minimises stigma, and acceptability of lay counsellors as delivery agents is established.[33] Inclusion of different school types and locations add to the potential generalisability of findings, and the use of validated and globally used instruments supports the generation of reliable and comparable data, including health economics data. Indian youth involvement, either as coresearchers, codesigners or in advisory panels, remains relatively novel. We will gain new insights into what works in this domain by supporting youth voice through advocacy via social media, filmmaking and pathways to policy impact. There are a number of study limitations. Study recruitment and engagement may be affected by school disruption from the COVID-19 pandemic, and the opt-in approach to consent may lead to participation bias. The universal approach with grade 9 students limits the extent to which interventions can be tailored to, for example, other mental health conditions or gender. We will be unable to identify adolescents with neurodevelopmental conditions or particular learning needs, meaning we will not know if they in particular can benefit from the interventions. Our school sampling strategy does not include representation of all possible Indian school structures and cultures.

## DISCUSSION

In India, a key contextual determinant of adolescent mental health is the nature of schooling.[13–22] This codesign and feasibility study is an important step towards an integrated, multicomponent whole-school approach to adolescent mental health in Indian secondary schools. School-delivered well-being programmes have the

potential to exceed the reach and scalability of mental healthcare systems in LMICs.[8–10] By targeting adolescent anxiety and depression, the interventions produced in SAMA will contribute to public mental health approaches to common adolescent mental health conditions in India, and, by promoting well-being factors, can inform health promotion strategies. Unique outcomes of the study include rich data on what Indian schools, and their communities, need and want from a whole-school approach to mental health as well insights into what aspects of international programmes need particular cultural adaptation. Mental health literacy among Indian adolescents, parents and teachers has been reported as low.[18–20] Our study will generate new knowledge on whether programmes developed in other countries, with likely different conceptualisations of mental health and responses to it, can be successfully coadapted to be acceptable and helpful in Indian schools. By adopting a randomised controlled waitlist design, the study will generate a solid evidence base for progression to a randomised controlled trial and inform understanding of the acceptability and rigour of data collection instruments and process evaluations in complex intervention studies in Indian schools. To date, school initiatives in India have mostly targeted physical health and life skills.[24–27] Drawing on the expertise of an international, interdisciplinary team, this study aims to contribute to the evidence base about what works in Indian schools to support adolescent mental health, and in particular, to reduce the prevalence and severity of adolescent anxiety and depression.

**Author affiliations**
[1]Psychology, University of Leeds Faculty of Medicine and Health, Leeds, UK
[2]Department of Psychiatric Social Work, National Institute of Mental Health and Neuro Sciences, Bangalore, Karnataka, India
[3]Health Economics Unit, School of Health and Population Sciences, University of Birmingham, Birmingham, UK
[4]School of Languages, Cultures and Societies, University of Leeds Faculty of Arts Humanities and Cultures, Leeds, UK
[5]Psychiatry, Department of Psychiatry, University of Oxford, Oxford, UK
[6]Improvement Academy, Bradford Institute for Health Research, Bradford, UK
[7]Sangath, Porvorim, Goa, India
[8]Global Health and Development, London School of Hygiene & Tropical Medicine, London, UK
[9]Indira Gandhi Medical College and Research Institute, Puducherry, Puducherry, India
[10]Leeds Institute of Health Sciences, University of Leeds, Leeds, UK
[11]The Barberry Centre for Mental Health, University of Birmingham, Birmingham, UK

**Contributors** SH-J is guarantor. SH-J, JN, PM, PB, MF, HA-J, PC, RMW and TM designed the study. PM and SH-J prepared the first draft of the manuscript with contribution from HA-J on health economics, from TM on policy pathways and PC on filmmaking. RMW assisted with writing the feasibility study design and data analysis. KH assisted with writing about implementation science. MF, PB and PK assisted with writing about PPIE, lay counsellors, ethics and safeguarding. SV assisted with writing about mental health literacy. All authors reviewed and approved the final draft.

**Funding** This work is supported by the UK's Medical Research Council, Economic and Social Research Council, National Institute of Health Research and UK Aid (grant number MR/T040238/1). All views expressed here are of the authors only.

**Competing interests** None declared.

**Patient and public involvement** Patients and/or the public were involved in the design, or conduct, or reporting, or dissemination plans of this research. Refer to the Method and analysis section for further details.

**Patient consent for publication** Not applicable.

**Provenance and peer review** Not commissioned; externally peer reviewed.

**ORCID iDs**
Siobhan Hugh-Jones http://orcid.org/0000-0002-5307-1203
Tolib Mirzoev http://orcid.org/0000-0003-2959-9187
Robert M West http://orcid.org/0000-0001-7305-3654

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
