## [Reviewer comments · BMJ Open]

ARTICLE DETAILS

TITLE (PROVISIONAL)	Safeguarding Adolescent Mental Health in India (SAMA): study protocol for co-design and feasibility study of a school systems intervention targeting adolescent anxiety and depression in India
AUTHORS	Hugh-Jones, Siobhan; Janardhana, N; Al-Janabi, Hareth; Bhola, Poornima; Cooke, Paul; Fazel, Mina; Hudson, Kristian; Khandeparkar, Prachi; Mirzoev, Tolib; Venkataraman, Surendran; West, Robert; Mallikarjun, Pavan

VERSION 1 – REVIEW

REVIEWER	Pelizza, Lorenzo Azienda USL di Parma, Department of mental Health and Pathological Addiction
REVIEW RETURNED	25-Aug-2021

GENERAL COMMENTS	Thanks for the opportunity to review this interesting paper on the SAMA program, a valuable protocol aimed to co-design and feasibility test a suite of multi-component interventions across the intersecting systems of adolescents, schools, families and their local communities in India. The description of the protocol is well-conducted, clear and comprehensive. The topic is crucial, especially for low and middle income countries. I have no suggestion. In my opinion, the paper is ready for publication.
--

REVIEWER	YAO, FEI Shanghai University of Traditional Chinese Medicine
REVIEW RETURNED	17-Oct-2021

GENERAL COMMENTS	The topic selection is innovative, and I hope it can provide some guidance for the psychological intervention of Indian adolescent. Good luck ! ! The problems are as follows: (1) The quality control of researchers and subjects should be described in more detail. (2) The intervention method should be appropriately described. (3) Is there a sample size calculation formula for sample size calculation? (4) Primary and secondary results should be more clearly stated. (5) Discussion should be made.
---

REVIEWER	Pellatt-Higgins, T University of Kent, Centre for Health Services studies
REVIEW RETURNED	22-Oct-2021

GENERAL COMMENTS	Can you mention in the introduction whether or not there is any ongoing research in this area, and if so how it relates to the SAMA study? At the bottom of page 6 you mention that there are additional benefits of having a small team of lay counsellors working in close partnership with the school, are you able to elaborate on this to justify the approach? Usually in feasibility work sample size calculations are not power based and the aim is to have enough participants to estimate key outcomes with a certain level of precision, can you provide rationale for using a power based approach? When a power based approach is used for definitive studies, it is usually based on 90% power and the 5% significance level, here you have used 98% power and a significance level of 1%. What is the rationale for this? Is the sample size based on other factors? Please can you elaborate on this and also whether the sample size takes account of the nested within participant statistical comparison? Please can you provide some justification for the sample size for the health economics evaluation? Otherwise, a very clearly written and comprehensive paper.
---

VERSION 1 – AUTHOR RESPONSE

Reviewer: 1

This reviewer felt the paper is ready for publication and no changes were requested.

Reviewer: 2

(1) The quality control of researchers and subjects should be described in more detail.

We hope we have interpreted your point correctly in assuming you are referring to the quality of lay counsellors who will be delivering the interventions and data collection assistant monitoring fidelity (rather than quality control of the research team or young people ('subjects')). We had reported on p10 that "Lay counsellors will receive full training prior to the launch of the feasibility study. Lay counsellors will complete weekly monitoring logs of intervention delivery in each school, including logging of modifications made (as per FRAME [48]) and will have weekly supervision by clinical team members. Fidelity monitoring will be undertaken by a team of trained data collection assistants." We also stated on p12 that "Lay counsellors will be trained and supported to optimise the intervention in situ by logging and overcoming implementation challenges as they arise, whilst keeping fidelity to intervention function." For clarity, we have now added to p10 that lay counsellors will "will be from the local areas, have at least a Masters qualification in psychology or a related discipline and experience in working with young people and / or schools. Prior to the launch of the feasibility study, they will receive full training over three months by a clinical psychologist on the team, supported by another team member experienced in training lay counsellors. Trainees will deliver multiple mock sessions of intervention delivery to support readiness for delivery in schools and quality control". We have also clarified that "Fidelity monitoring will be undertaken by a team of trained data collection assistants who have been heavily involved in the intervention development stage and will be highly familiar with the implementation protocol. Fidelity monitoring reports will be discussed frequently with the research manager to identify needs for lay counsellor action / support or intervention modification (see WP5)."

(2) The intervention method should be appropriately described.

Project SAMA adopts a co-design process and therefore the final interventions, and their exact implementation methods, cannot be specified at this point since these are to be produced and agreed with stakeholders. At this point, we can only specify the four intervention aims and that delivery will be via lay counsellors. When finalised, we will publish the intervention manuals on the project website and publish, with outcomes, in peer reviewed journals at project end. Thus, we anticipate complete transparency in intervention methods over time.

(3) Is there a sample size calculation formula for sample size calculation?

We do not report this in the paper (as this is not standard) but we can clarify here that we followed the power calculation formula as per the `pwr.t2n.test` from the R library `pwr` version 1.3-0, informed by Stephane Champely (2020). `pwr: Basic Functions for Power Analysis`. R package version 1.3-0. <https://CRAN.R-project.org/package=pwrS>.

(4) Primary and secondary results should be more clearly stated.

As this is a study protocol, we have no results to report at this stage. The primary and secondary outcome measures that are planned are reported in Table 1. It is possible that, through co-design processes, new secondary outcomes measures could be included. For clarity, we have added the following statement on p6: “It is possible that co-design discussions may lead to the inclusion of additional outcome measures.”

(5) Discussion should be made.

As this is a study protocol, we have no results to discuss at this point. The paper follows the structure for protocols advised by the journal.

Reviewer: 3

(1) Can you mention in the introduction whether or not there is any ongoing research in this area, and if so how it relates to the SAMA study?

We explain in the introduction that the majority of the work in India to date has focused on physical and life skills programs, with just a few focused on mental health. Based on systematic reviews (Barry et al., 2013; Bradshaw et al, 2021; Fazel et al 2014), we described the evidence from these as “promising but limited”. We also reported on p7 the specific work in India. However, we agree with the reviewer that we could be clearer about how SAMA relates to these and have added on p5:

One did not report any beneficial clinical outcomes [24] and one has yet to establish effectiveness. [32] indicating that life-skills interventions may be insufficient to impact adolescent anxiety and depression. The third is a health promotion programme (SEHER) focusing on school climate, and targeting physical and sexual health, bullying, gender equality and depressive symptoms. It focused on promoting adolescents’ social and problem-solving skills, engaging adolescents, teachers, and parents in school-level decision-making and delivering factual knowledge about health and risk behaviours to the school community. [26]

Our study **Safeguarding Adolescent Mental heAlth in India (SAMA)** builds on learning from SEHER in terms of the value and acceptability of whole school, multi-component interventions in India schools and the contribution of school climate to this. Project SAMA extends SEHER in several ways. Our health objective is to reduce symptoms of both anxiety and depression in symptomatic adolescents and to promote the wellbeing all school-going adolescents. We will co-design, and feasibility test, a suite of school interventions in India targeting multiple risks and protective factors across the school system. We extend SEHER by including a co-designed interventions targeting teachers and parents, as well as an adolescent mental health psychoeducation intervention. We also augment the school climate intervention of SEHER via co-production. The SAMA study will take place between January 2021 and December 2023.

(2) At the bottom of page 6 you mention that there are additional benefits of having a small team of lay counsellors working in close partnership with the school, are you able to elaborate on this to justify the approach?

The World Health Organization endorses task-shifting in low-resource settings. This involves assigning tasks and responsibilities from more specialised to less specialised groups. It is recognised that task-shifting requires clearly defined roles and effective systems of supervision and referral. We have now added a comment to this effect this on p6, when we first introduce lay counsellors.

We had cited two studies in the manuscript to justify our use of lay counsellors.

- Rajaraman et al. (2012) who established the feasibility and effectiveness of lay counsellor to deliver a school health and life skills programmes in Goa.
- Shinde et al. (2018) provide the strongest support for lay counsellors. They conducted a cluster-randomised trial to assess the effectiveness of a multi-component whole-school health promotion intervention (SEHER) in Bihar, India. The study examined SEHER intervention delivered by a lay counsellor vs a teacher vs control in 75 schools. They found that SEHER had substantial beneficial effects on school climate and health-related outcomes when delivered by lay counsellors, but no effects when delivered by teachers. **For clarity, we have added to page 6** “Notably, the cluster randomised trial of SEHER (a multicomponent school health promotion program) conducted across 75 schools in Goa was found to be effective when delivered by lay counsellors but not teachers [26].

Our claim about the additional benefits of lay counselors working in school is informed by reports of SEHER (Shinde et al., 2018) and the experience of a Co-I (PK) who delivers SEHER. We have added a sentence to this effect on p6 These [benefits] include each lay counsellors having a co-counsellor to work alongside to deliver SAMA, counsellors being able to get to know particular schools well (i.e. their systems, culture), having constancy in the lay counsellors so school staff and students can build familiarity and trust, and having a defined conduit for communication about SAMA within schools and between school and the study team.

(3) Usually in feasibility work sample size calculations are not power based and the aim is to have enough participants to estimate key outcomes with a certain level of precision, can you provide rationale for using a power based approach?

We use a power-based approach to permit sensitivity testing of our candidate outcome measures for WP1. First, to secure an acceptable sample size for our smallest group of participants (teachers) for a feasibility study (n=30 recommended), we need to recruit 8 schools. From 8 schools, we can then recruit approx. n= 960 Grade 9 students for WP2 (as we are delivering to whole year groups). We

then calculated what power we could achieve from this sample to test our WP1 candidate primary outcome measures. We have now added more detail on p11 to explain this.

(4) When a power based approach is used for definitive studies, it is usually based on 90% power and the 5% significance level, here you have used 98% power and a significance level of 1%. What is the rationale for this? Is the sample size based on other factors? Please can you elaborate on this and also whether the sample size takes account of the nested within participant statistical comparison?

In addition the response to (3), we have now added further clarification on this to p12: "There may, however, be differences in the operation of the intervention between subgroups, such as government versus private schools. In that case, assuming an even division of samples, the power becomes 85% with 80% recruitment and 78% with 65% recruitment. A significance level of 1% is used rather than 5% since the sensitivity of the outcome measures is sought to be compared – that is, the (SDQ) [49] will be compared in sensitivity, in this setting with the (RACDS) [50]."

(5) Please can you provide some justification for the sample size for the health economics evaluation?

The health economics analysis will involve quantitative and qualitative feasibility testing of quality-of-life instruments. The quantitative sample size was not selected to estimate the significance of a specific outcome, but instead, to balance the desire to not over-burden individuals, while still providing a dataset where descriptive judgements could be made about completion rates of instruments and items. The lower sample size for the qualitative component reflects the much 'richer' data generated through qualitative interviews and the more in-depth coding and thematic analysis. We have added a comment to this effect on p12.

VERSION 2 – REVIEW

REVIEWER	YAO, FEI Shanghai University of Traditional Chinese Medicine
REVIEW RETURNED	11-Dec-2021

GENERAL COMMENTS	Thank you for giving me this chance to review this manuscript again. This is a huge and laborious research. Good luck to the researchers Some comments for improvement and understanding this manuscript. (1) There should be a special paragraph on the advantages and limitation of this research. (2) It is recommended to add a brief discussion at the end
---

REVIEWER	Pellatt-Higgins, T University of Kent, Centre for Health Services studies
REVIEW RETURNED	16-Dec-2021

GENERAL COMMENTS	Very clearly written and well designed feasibility study, all reviewers comments have been addressed comprehensively and I have no further comments.
--

VERSION 2 – AUTHOR RESPONSE

Thank you to the reviewers. We have now added a paragraph on the strengths and limitations of the study as well as a brief discussion.